# NF-κB-Dependent and -Independent (Moonlighting) IκBα Functions in Differentiation and Cancer

**DOI:** 10.3390/biomedicines9091278

**Published:** 2021-09-21

**Authors:** Lluís Espinosa, Laura Marruecos

**Affiliations:** Cancer Research Program, Institut Mar d’Investigacions Mèdiques, CIBERONC, Hospital del Mar, Doctor Aiguader 88, 08003 Barcelona, Spain; lmarruecos@imim.es

**Keywords:** IκBα, SUMOylation, NF-κB, polycomb repression complex (PRC) 2, cancer

## Abstract

IκBα is considered to play an almost exclusive role as inhibitor of the NF-κB signaling pathway. However, previous results have demonstrated that SUMOylation imposes a distinct subcellular distribution, regulation, NF-κB-binding affinity and function to the IκBα protein. In this review we discuss the main alterations of IκBα found in cancer and whether they are (most likely) associated with NF-κB-dependent or NF-κB-independent (moonlighting) activities of the protein.

## 1. Introduction

IκBα is considered to play an almost exclusive role as inhibitor of the NF-κB signaling pathway. However, previous results have demonstrated that SUMOylation imposes a distinct subcellular distribution, regulation, NF-κB-binding affinity and function to the IκBα protein. In this review we discuss the main alterations of IκBα found in cancer and whether they are (most likely) associated with NF-κB-dependent (canonical) or NF-κB-independent (moonlighting) activities of the protein.

## 2. Altered IκBα Activity in Cancer

NF-κB signaling plays an essential role as regulator of the immune system and inflammation. In addition, it contributes to several aspects of tumorigenesis by direct modulation of cellular functions such as proliferation, apoptosis inhibition or cellular migration. The mammalian NF-κB family of proteins is composed of five transcription factors (TF), RelA (p65), RelB, c-Rel, p105/p50 (NF-κB1) and p100/p52 (NF-κB2), that can induce gene transcription as combinations of homo- and heterodimers, the p50/p65 heterodimer being the most prevalent. Under basal conditions, NF-κB TFs are maintained in an inactive state in association with the inhibitors of kappaB (IκB), which impose NF-κB cytoplasmic retention. Canonical NF-κB activation induced by PAMPs (Pathogen-associated molecular patterns) and pro-inflammatory cytokines is initiated by the IKK kinase complex that phosphorylates IκB thus inducing K48-linked ubiquitination at K21 of IκB and proteasomal degradation, leading to release of NF-κB TFs that then translocate to the nucleus to activate specific gene transcription [1] (Figure 1). Alternative modifications of IκB have also been shown to affect NF-κB activity under specific conditions. In hypoxia, which is important in the hypoxic tumor environment, SUMOylation of IκBα at K21 prevents its ubiquitination and degradation, but facilitates p65 release and NF-κB activation [2]. SUMOylation of IκB had initially been described associated with non-phosphorylated IκBα forms [3]. Other stimuli, such as H_2_O_2_, EGF (Epidermal Growth Factor) or pervanadate, induce tyrosine phosphorylation of IκBα (at Y42) that leads to NF-κB activation both independent or dependent on IκBα degradation [4,5,6,7,8].

Constitutive activation of the NF-κB pathway has been identified in multiple solid and hematopoietic tumors by either detection of nuclear NF-κB factors or by presence of NF-κB-related transcriptional patterns. Several mechanisms including gene amplification or overexpression, chromosomal rearrangement, mutations, truncations and splicing variants have been found to impose constitutive NF-κB activation in cancer. In some cases, elevated NF-κB activity is associated with defective IκB function [9,10].

## 3. IκBα Activation and Functions

The IκB proteins, including IκBα, IκBβ, IκBε, IκBζ, Bcl-3 (B-cell lymphoma 3-encoded protein), and the precursor Rel proteins p100 (IκBδ) and p105 (IκBγ), are structurally characterized by the presence of multiple ankyrin repeats, which mediate protein–protein interaction and cytoplasmic NF-κB retention. Although this family of proteins is primarily known from its almost exclusive role in canonical NF-κB inhibition [1], several NF-κB-independent functions for specific IκB subunits have been identified. Initially, it was demonstrated that physical association of IκBα to histone deacetylases (HDACs) increase NF-κB-independent transcription by cytoplasmic retention of HDAC1 and HDAC3 [11]. IκBα can also bind the chromatin at specific genomic regions to regulate gene transcription by modulation of the chromatin editing polycomb repression complex (PRC) 2 [12,13]. Mechanistically, phosphorylated and SUMOylated IκBα interacts with chromatin by direct binding to the acetylated N-terminal tail of H4 [13]. The association of IκBα to specific genes facilitates PRC2 recruitment under basal conditions but imposes the capacity of activation in response to inflammatory cytokines such as TNFα or following differentiation [13,14] (Figure 1). In the same direction, it has recently been demonstrated that the dynamics of NF-κB activation can reprogram the epigenome in a stimulus specific manner [15], which may be linked to changes in IκBα protein levels.

Transcriptional regulation mediated by chromatin-bound IκBα affects about 10% of all PRC2 target genes in the different models studied, and involves genes related with development, stemness and tissue homeostasis [13,14,16]. Importantly, IκBα deficiency leads to defective maturation of tissue stem cells, which are then retained in a fetal state [13,14]. The nuclear function of IκBα was first described in mammalian cells and tissues including skin [13] and intestine [14], but it is already present in organisms like *Drosophila melanogaster* [13] and *Caenorhabditis elegans* [16]. Notably, *Caenorhabditis elegans*, as the rest of nematodes, lacks recognizable NF-κB factors, strongly suggesting that nuclear and polycomb-related IκBα functions appeared in the evolution before or in parallel to its role as NF-κB inhibitor.

Therefore, there is cumulative data indicating that alterations related with IκBα function not only affect NF-κB pathway but IκBα exerts moonlighting functions including regulation of PRC2 activity on specific gene sets, which are pivotal for cancer initiation and progression.

## 4. IκBα in Hematologic Diseases

The first mutations of *NFKBIA*, the gene codifying for IκBα, in cancer were detected in Hodgkin lymphoma (HL), a B-cell lymphoma developed in lymph nodes, characterized by the presence of giant Reed-Sternberg (RS) cells. Several inactivating mutations (e.g., small insertions, deletions or nonsense mutations) in this gene are the direct cause of constitutive activation of the NF-κB pathway observed in RS cells. These mutations produce non-functional proteins, which usually lack fragments of the ankyrin domain and/or the C-terminal region. About 37% of HL patients show *NFKBIA* mutations, with more than 18% of them having loss-of-function mutations (about 10% in both alleles) [17,18,19,20,21].

Mutations in *NFKBIA* were also found in around 22% of patients with Gray zone lymphoma (GZL), a rare hematological disease with features intermediate between large B-cell lymphoma (LBCL) and classical HL. These mutations (non-sense or frameshift) are among the most prevalent in GZL patients [22]

A high presence of *NFKBIA* mutations has also been reported in diffuse large B-cell lymphoma (DCLBL) patients [23] and in the RC-K8 DCLBL cell line [24], which result in the total absence of the protein. Additional alterations, such as gene fusions involving *RELA* have been found to impose constitutive NF-κB activation in these cells [24,25]. Notably, high levels of NF-κB signaling involving nuclear c-Rel have also been observed in mediastinal large B-cell Lymphoma (MLBCL), a subtype of DCLBL with clinical and molecular features that resemble the ones observed in HL patients [26], which lack mutations in the coding region of NFKBIA [27].

In addition to its linkage with B-cell neoplasms, aberrant IκBα function was also detected in chronic myeloid leukemia (CML), a myeloproliferative disorder driven by the BCR-ABL translocation. In this type of cancer, IκBα is expressed at high levels and associated with the BCR-ABL and p53 proteins. This complex imposes cytoplasmic retention of p53 thus precluding its nuclear activity as tumor-suppressor [28]. A crosstalk between IκBα and NF-κB with the tumor-suppressor p53 has also recently been identified and characterized in other oncogenic alterations such as tissue-specific lymphomas arising in the *Trp53*-deficient mouse model [29] and acute lymphoblastic leukemia cells [30].

In ectodermal dysplasia, mutations leading to N-terminal truncation of IκBα leading to a non-degradable protein with higher inhibitory activity on NF-κB have been identified [31] but the mechanisms linking this particular alteration with disease remains unexplored.

Moreover, different hematological diseases have been linked to the presence of polymorphisms in the *NFKBIA* gene likely associated with altered protein activity or gene expression. For example, polymorphisms localized in regions that may affect the rate of protein degradation are more frequent in multiple myeloma patients than in healthy subjects [32,33]. Some of these polymorphisms are also found in lymphoma [18,19], further supporting their functional relevance.

In general, whether *NFKBIA* mutations promote cancer initiation and/or progression through activation of NF-κB signaling or by alternative NF-κB-independent mechanisms in the different systems has not directly been analyzed.

## 5. IκBα Loss in Glioma

It has consistently been observed that NF-κB activity is significantly higher in brain tumors than in normal tissues [34,35]. Notably, analysis of a large set of grade IV glioblastoma patients demonstrated reduced expression of IκBα in these tumors. Reduced IκBα levels in glioblastoma was not linked to mutations in the coding or promoter regions of the *NFKBIA* gene, but to gene copy number alterations. Monoallelic deletions were detected at low frequency in classical glioblastoma (~6%) and were more common in non-classical subtypes (~25%). Notably, heterozygous *NFKBIA* deletion and *EGFR* amplifications are mutually exclusive suggesting that both alterations impact in the same pathway, which should be further investigated. *NFKBIA* deletions and reduced IκBα levels are significantly associated to unfavorable outcomes and tumor recurrence in patients [36,37,38]. In vitro experiments using glioblastoma cell lines demonstrated that the reintroduction of *NFKBIA* gene or IκBα protein by nanoparticles induced apoptosis and increased the sensitivity of tumor cells to chemotherapy [39]. In addition to allelic loses, a single-nucleotide polymorphism in exon 1 *of NFKBIA*, which leads to reduced expression of IκBα, is more frequently present in glioblastoma patients than in the healthy population and is associated to poor prognosis [40].

In lower-grade glioma (LGGs), which is a heterogeneous group of brain tumors that can progressively evolve into higher-grade gliomas, heterozygous *NFKBIA* deletions are observed in approximately 7% of patients, being more common in the more advanced stage III tumors (~11%) than in stage II tumors (~3%). Moreover, it was demonstrated that both NFKBIA deletion and reduced levels of IκBα are associated with tumor aggressiveness and poor prognosis. In fact, the dosage of *NFKBIA* was identified as an independent factor for 5-year survival and recurrence-free survival [37].

## 6. IκBα Alterations in Other Solid Tumors

Multiple evidences indicate a pro-tumorigenic activity of constitutive or aberrant NF-κB signaling in several solid tumors including skin [41], prostate [42,43], breast [44,45,46], stomach [47], intestine [48,49] and lung [50].

Despite the high frequency of increased NF-κB activity in solid tumors, alterations in IκBα function have marginally been identified. In lung cancer patients, loss of heterozygosity (LOH) in 14q, where the *NFKBIA* gene is localized, was previously identified [51]. A different study demonstrated that about 16% of never-smoker patients with non-small cell lung carcinoma (NSCLC) showed a loss of IκBα expression. IκBα deficiency was more frequent in patients without any other driven genetic alterations such as EGFR, K-Ras or EML4-ALK, pointing out *NFKBIA* silencing as a driving force in the development of this subset of tumors [52]. Whole genome sequencing of one NSCLC patient suggested that inactivating mutation in *NFKBIA* may support defective IκBα expression [53]. In addition, EGF can induce constitutive NF-κB activity in NSCLC cells by Tyrosine 42 phosphorylation of IκBα, which leads to its ubiquitin-independent degradation [6]. Importantly, IκBα silencing by interference RNA expression was found to rescue EGFR-mutant lung cancer cells from erlotinib (EGFR-tyrosine kinase inhibitor) treatment. In agreement with this observation, high levels of IκBα predicts a therapeutic response in patients treated with EGFR inhibitors [54]. In contrast, compound targeting IκBα enhanced the susceptibility of lung cancer cells to cisplatin, leading to reactive oxygen species (ROS)-induced cell death [55].

The *NFKIA* gene was found among the most frequently mutated genes in whole-exome sequencing analysis of nasopharyngeal carcinoma patients. The identified mutations include missense single nucleotide variations or insertions/deletions that result in stop codon gain or changes in frameshift, especially in the ankyrin repeat region. These loss-of-function mutations lead to protein truncation that induce altered NF-κB response. Loss of 14q region, containing the *NFKIA* locus, was also observed in a high proportion of nasopharyngeal carcinoma patients [5,56,57,58,59,60]. Interestingly, not only *NFKIA* but also *NFKBIB* gene was found to be silenced in most nasopharyngeal carcinoma tumors leading to increased NF-κB signaling [61].

Polymorphisms in the *NFKIA* gene also play an important role in different solid tumors, although the association between *NFKIA* polymorphisms and cancer susceptibility varies among the different populations and tumor types. For example, a polymorphism (A > G) in the 3′UTR of *NFKBIA* and insertions/deletions in the promoter region were found to increase the predisposition to colorectal cancer [62,63] and was also linked to poorer survival rate and Crohn’s disease [62]. Other polymorphisms have been found that increase the risk of gastric cancer [64], ovarian [65] or melanoma [66], and polymorphisms in the promoter region of *NFKIA* were associated with increased risk of HBV (hepatitis B virus)-induced hepatocellular carcinoma (HCC) [67] and prostate cancer in the Chinese population [68]. Moreover, in this type of cancer, decreased IκBα expression levels are associated with higher tumor grade [69].

Aberrant IκBα function was also identified in skin cancer. In particular, nuclear IκBα, which is robustly detected in non-transformed basal-layer keratinocytes, is lost in transformed cells (mutant H-RasV12 plus ΔNp63α) as well as in samples from patients with aggressive squamous cell carcinoma (SCC). Moreover, loss of nuclear IκBα and cytoplasmic accumulation was progressively acquired with tumor malignization [13]. In this sense, non-invasive lesions such as actinic keratosis or Bowen’s disease tumors still retain nuclear IκBα that is completely absent from SCC, as mentioned. Mechanistically, loss of nuclear IκBα led to the release of PRC2 complex from chromatin and the subsequent transcription activation of target genes including *HOX* homologues. *HOX* genes deregulation is a common trait in cancer (reviewed in [70,71]).

The relevance of IκBα in skin homeostasis and tumorigenesis was repeatedly demonstrated in different mouse models. In 1999, Van Holgerliden and colleagues generated mouse lines expressing the IκBα_S32,36A_ super-repressor (SR) mutant in basal-layer keratinocytes. Unexpectedly, these mice, which display constitutive NF-κB inhibition, developed hyperplasia and spontaneous SCC within 3 months [72]. A comparable phenotype was observed in a mouse carrying IκBα_S32,36A_ that contains an additional stabilizing deletion of the COOH-terminal PEST sequence [73]. Moreover, co-expression of IκBα-SR with mutant *KRAS* (G12V) increased tumor invasiveness [74]. Together these results indicate that increased tumorigenesis associated to altered IκBα function in the skin is not due to increased NF-κB signaling and points to alternative PRC2-related functions.

Nuclear IκBα was also robustly detected in the intestinal stem cell compartment. Indicative of its functional relevance, mice deficient on IκBα showed fetal traits in the intestinal stem cell population and increased regeneration capacity after intestinal damage [14]. Based on the mechanistic similarities between regeneration and cancer, it is likely to speculate that IκBα may also play a relevant role in intestinal tumor initiation and/or progression. Recently, it was found that intestinal differentiation is linked to histone clipping, in particular of histone H3 [75] and H4 [76]. Loss of the N-terminal tail of H4 by action of trypsin and chymotrypsin leads to the release of IκBα from the chromatin, being dynamic chromatin binding/dissociation of IκBα required for proper differentiation of intestinal lineages. We propose that both IκBα and H4 clipping might cooperatively control intestinal homeostasis and oncogenic transformation. Interestingly, mutations in histone H4 have been described in numerous types of cancer such as endometrial, bladder, head and neck, esophagogastric or colorectal carcinomas. Importantly, some of the identified mutations are contained in the region of cleavage, in particular between K16 and K20 residues (reviewed in [77,78]).

## 7. Targeting NF-κB or/and Chromatin Editing Enzymes for Treating IκBα-Deficient Tumors

NF-κB has been considered as a targetable pathway in cancer for many years. However, general inhibition of NF-κB signaling was found to be extremely toxic due to its widespread effects (reviewed in [79]). One of the first inhibitors targeting NF-κB that was approved by its use in multiple myeloma was Bortezomib, a reversible proteasome inhibitor that prevents IκBα degradation [80]. Later on, it was shown that general NF-κB inhibitors led to massive inflammation in mice by increasing IL1-β secretion and inflammasome activation [81], thus questioning the suitability of these inhibitors for cancer therapy. One possibility, which should be tested, would involve using short-duration treatments or treatments involving suboptimal NF-κB inhibitors dosage. Alternatively, specific tumors carrying defective IκBα activity but normal NF-κB activity may display chromatin alterations (due to defective moonlighting IκBα functions) that could be therapeutically targeted with PRC2 inhibitors that are currently under investigation [82]. Further research on IκBα-deficient tumors will provide valuable information on novel druggable targets, other than NF-κB, that will benefit treatment of a specific group of cancer patients.

## 8. Conclusions

Together, published data linking altered IκBα and cancer indicate that IκBα exerts an essential role not only in tissue homeostasis, but also in the initiation, maintenance and progression of several types of cancer (Table 1). Moreover, altered IκBα activity leading to cancer involves both NF-κB-dependent and NF-κB-independent (moonlighting) functions. We propose that IκBα alterations that impose constitutive activation of NF-κB may increase proliferation and invasion, and prevent cell apoptosis through transcriptional activation of specific NF-κB target genes (i.e., *IL6*, *CyclinD1*, *cIAP*). In contrast, altered nuclear IκBα activity may either inhibit or potentiate PRC2-mediated regulation of stem cell- or differentiation-related transcription thus precluding physiologic gene regulation modulated by signals provided by the tumor stroma or the stem cell niche (i.e., cytokines or growth factors) (Figure 2). This possibility is also in agreement with the altered intestinal stem cell regulation and tissue homeostasis produced after deletion of the PRC2 elements [83,84]. Therefore, maintenance of stem cells in an immature state linked to impaired differentiation may facilitate cell transformation and tumor progression. We propose that not only NF-κB inhibitors but drugs targeting chromatin-editing enzymes might be tested in combination therapies for treating patients carrying IκBα-defective tumors.

## Figures and Tables

**Figure 1 biomedicines-09-01278-f001:**
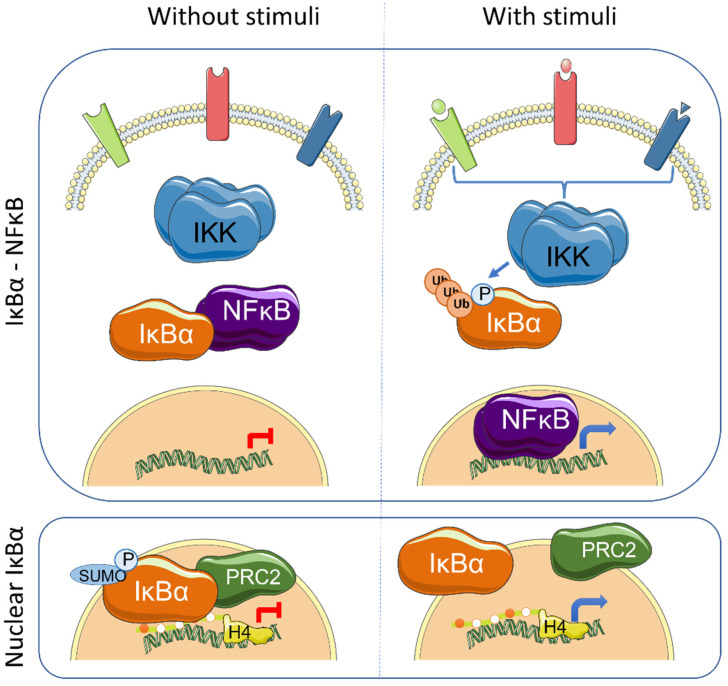
Graphical representation of the main IκBα-regulated pathways: NF-κB and nuclear SUMOylated IκBα. Left panels show the status of the involved elements under unstimulated conditions whereas right panels indicate changes produced after cytokine induction.

**Figure 2 biomedicines-09-01278-f002:**
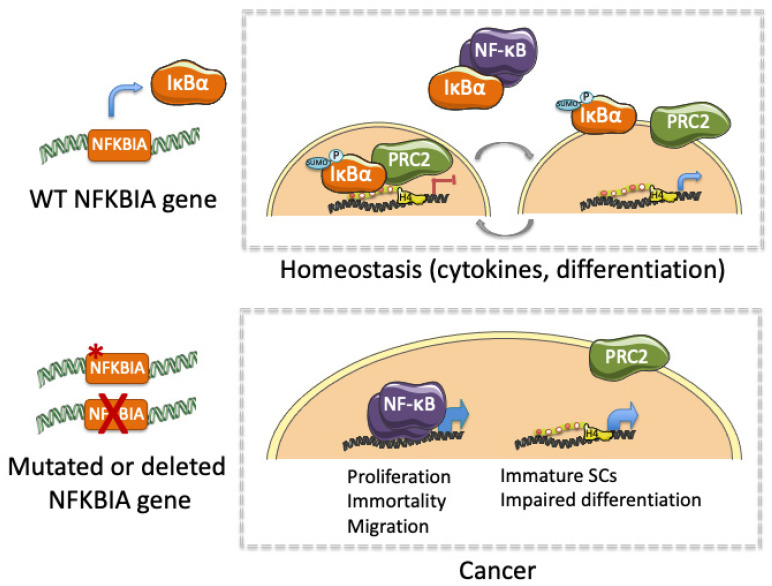
Graphical representation of the functional impact of IκBα under physiological conditions (homeostasis) and following genetic or functional alteration (cancer).

**Table 1 biomedicines-09-01278-t001:** Table representing the alterations of *NFKBI* gene and/or IκBα activity associated with different types of human cancer and murine cancer models.

Tissue	Cancer	IκBα Alteration	Effects	References
Hematopoietic	Hodgkin lymphoma (HL)	Inactivating mutations that produce non-functional protein	NF-κB constitutive activation	[16,17,18,19,20]
	Gray zone lymphoma (GZL)	Inactivating mutations	No evaluated	[22]
	B-cell lymphoma (DCLBL)	Inactivating mutations with absence of the protein	NF-κB constitutive activation	[22,23]
	Chronic myeloid leukemia (CML)	High expression	Complex with BCR-ABL and p53, avoiding its nuclear activity as tumor-suppressor	[24]
Brain	Glioblastoma	Reduced gene copy number	No evaluated	[27,28,36]
	Low-grade glioma	Reduced gene copy number	No evaluated	[37]
Lung	Non-small cell lung carcinoma (NSCLC)	Loss of heterozygosity (LOH) or inactivating mutations	No evaluated	[39,40]
Head and neck	Nasopharyngeal carcinoma	Inactivating mutations and loss of 14q region	NF-κB constitutive activation	[5,56,57,58,59,60]
Skin	Squamous cell carcinoma (SCC)	Loss of nuclear IκBα and cytoplasmic accumulation	Altered IκBα /PRC2 target gene transcription	[12]
		IκBα super-repressor (SR) mutant	Constitutive NF-κB inhibition	[47,48]
		IκBα super-repressor (SR) mutant + mutant *KRAS* (G12V)	Invasiveness of tumor	[50]

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
