# Peer review of "NF-κB-Dependent and -Independent (Moonlighting) IκBα Functions in Differentiation and Cancer"

_biomedicines, 2021, doi:10.3390/biomedicines9091278_

Round 1
Reviewer 1 Report
In this review Espinosa and Marruecos summarize how IkBa impacts upon cancer development both through its role as a key modulator of NF-kB signaling and through its NF-kB-independent nuclear functions. It provides a concise and informative up-to-date description of an important field of research especially useful for non-experts.
The interest of this review is that it summarizes for NF-kB experts or non-experts specific functions of IkBa not related to IKK activation that should deserve more consideration.
This review describes both NF-kB-dependent and -independent ("moonlighting") functions of IkBa but sorting out the respective contributions of these two modes of action in pathophysiology (See Fig. 2*) remains a challenge. Therefore, the title may be a bit misleading since it only points to NF-kB-independent functions. It could be changed to :
"NF-kB-Dependent and Moonlighting Functions of IkBa in Differentiation and Cancer".
*By the way, Fig. 1 should be table 1 and Fig. 2 should be Fig. 1.
Author Response
In this review Espinosa and Marruecos summarize how IκBα impacts upon cancer development both through its role as a key modulator of NF-κB signaling and through its NF-κB-independent nuclear functions. It provides a concise and informative up-to-date description of an important field of research especially useful for non-experts. The interest of this review is that it summarizes, for NF-κB experts or non-experts, specific functions of IκBα not related to IKK activation that should deserve more consideration.
Answer: We thank the reviewer the positive assessment of our work.
Reviewer: This review describes both NF-κB-dependent and -independent ("moonlighting") functions of IκBα but sorting out the respective contributions of these two modes of action in pathophysiology (See Fig. 2*) remains a challenge. Therefore, the title may be a bit misleading since it only points to NF-κB-independent functions. It could be changed to "NF-κB-Dependent and Moonlighting Functions of IκBα in Differentiation and Cancer".
Answer: We have changed the title as suggested by reviewer 1.
Reviewer: *By the way, Fig. 1 should be table 1 and Fig. 2 should be Fig. 1.
Answer: We have changed it.

Reviewer 2 Report
The current review by Espinosa and Marruecos is very informative about the I-kappa-B-alpha (IkBa) biology. Indeed this novel transcriptional inhibitor acts not only to keep the function of NFkB in control allowing normal homeostasis to prevail but also impact several independent functions.
However, notably the review lacks any take home message for the readers. Many of the published works and their significance to cancer indications are well known. So where do we go from there ?
Are there new and upcoming strategies to regulate IkBa biology? can we show in cell lines at least gene editing by Crispr rescue cancer cells to normal phenotype? or are there anyways the authors suggest that IkBa or IkBa-NfKb complex can be targeted by small molecules or siRNAs.
Secondly the review lacks clarity in terms of reading. It becomes very helpful for readers when complex signal transduction pathways are shown as schematic diagrams which is largely absent except for the last figure 2.
Including the recommended changes will make the review attractive to a much larger reader base and potentially suitable for review and consideration for publication in Biomedicines.
Author Response
The current review by Espinosa and Marruecos is very informative about the I-kappa-B-alpha (IκBα) biology. Indeed, this novel transcriptional inhibitor acts not only to keep the function of NF-κB in control allowing normal homeostasis to prevail but also impact several independent functions.
Reviewer: However, notably the review lacks any take home message for the readers. Many of the published works and their significance to cancer indications are well known. So where do we go from there?
Answer: We have included a take home message at the end of the conclusions section.
Reviewer: Are there new and upcoming strategies to regulate IκBα biology? can we show in cell lines at least gene editing by Crispr rescue cancer cells to normal phenotype? or are there anyways the authors suggest that IκBα or IκBα-NF-κB complex can be targeted by small molecules or siRNAs.
Answer: We do not suggest (or even think) that rescuing canonical/NF-κB-related IκBα activity by using the IκBα-SR construct should revert tumor cells to normal physiology, instead be mention in the review that mice carrying IκBαS32,36A display constitutive NF-κB inhibition but they developed hyperplasia and spontaneous SCC within 3 months, indicating that increased tumorigenesis associated to altered IκBα function (at least in the skin) is not due to increase NF-κB signaling and point out to alternative PRC2-related functions. Thus, more complex strategies should be designed for treating IκBα-deficient tumors, which is now indicated in the conclusions section.
Secondly the review lacks clarity in terms of reading. It becomes very helpful for readers when complex signal transduction pathways are shown as schematic diagrams which is largely absent except for the last figure 2.
Answer: We are including a scheme in new figure 1 representing the signaling pathways that regulate canonical and alternative IκBα functions.
Reviewer: Including the recommended changes will make the review attractive to a much larger reader base and potentially suitable for review and consideration for publication in Biomedicines.
Answer: We have included reviewers’ suggestions.

Round 2
Reviewer 2 Report
The authors have addressed several critical issues of the review including clarifying language and figures . Should be accepted to Biomedicines in its present form.
This manuscript is a resubmission of an earlier submission. The following is a list of the peer review reports and author responses from that submission.